# Mother-child autonomic nervous system interaction as an indication of parental stress: 24-hour cross recurrence plot analysis

Jiarui Li[1]*, Michiko Matsunaga[2], Masako Myowa[2], Yukie Nagai[1]

1 International Research Center for Neurointelligence (WPI-IRCN), The University of Tokyo, Tokyo, Japan,
2 Graduate School of Education, Kyoto University, Kyoto, Japan

* ljr10225008@gmail.com

## Abstract

Mothers and children interact through behaviors and mutual physiological signals. Recent research has suggested that physiological interactions are sensitive biomarkers for parenting challenges such as childcare stress. Most studies have focused on short-term, activity-based interactions. However, few studies have examined longer timescale data, such as interactions containing circadian rhythms, which are closely associated with stress. This study aimed to explore the link between childcare stress and mother-child autonomic nervous system activities in daily interactions using 24-hour at-home data. Cross-recurrence plots were used to quantify interactive patterns to reflect the rich nonlinear dynamics of autonomic nervous system signals and the complex interplay between mothers and children. The relationships between quantified low-dimensional cross-recurrence plot representations and parenting stress were evaluated through multiple regression analysis. The cross-recurrence plots effectively quantified complex, nonlinear, day-to-night interactions of mother-child autonomous nervous system activities and significantly predicted maternal parenting stress, offering novel insights into daily parental stress dynamics beyond recurrence plots that merely quantify the activities from individual systems.

## Introduction

Incongruity in the psychosomatic states of children and their caregivers contributes to parenting problems such as caregiver stress during daily interactions. Mothers and children are close interactors during daily caregiving life, so their physiological and behavioral signals often present interactive patterns. For example, research has examined the coordination of behavioral cues, such as facial expressions, vocalizations, and gaze orientation [1], in controlled and natural settings, ranging from laboratory experiments to at-home studies during mother-child interactions. Additionally, recent advances in vital sensing technology have made it possible to collect everyday

**Data availability statement:** Data for the analysis of this paper is available at https://osf.io/d6phj/.

**Funding:** This study was supported by a Grant-in-Aid for Scientific Research from the Japan Society for the Promotion of Science (JSPS) (17H01016, 19K21813, and 21H04981 to M. Myo., which funded for data collection, and the decision to publish; 22K20314 and 24K16785 to J.L., which funded for data analysis, and preparation of the manuscript); A Grant-in-Aid for JSPS Fellows (19J15173 and 22J01448 to M. M., which funded the data collection and data analysis); A grant from the Center of Innovation Program, Japan Science and Technology Agency (JPMJCE1307 to M. Myo., which funded data collection and data analysis; JPMJCR21P4 to Y. N., which funded the decision to publish); A grant from the Moonshot R&D Program, Japan Science and Technology Agency (JPMJMS2296, JPMJMS2307 to M. Myo., which funded for data collection; JPMJMS2292 to Y. N., which funded for data analysis); A grant from the Next Generation Artificial Intelligence Research Center of The University of Tokyo ("Circadian Patterns Analysis on Autonomic Nervous System (ANS) data of Infants and Their Caregivers" to J.L., which funded for data analysis); A donation from the Hama Gakuen Corporation and the Shoei Kosan Corporation ("Childhood Dietary Habits and Gut Microbiome" to M. Myo.); and a donation from the Chouyoukai ("Dietary Habits and Intestinal Microbiota during Infancy" to M. Myo.). These donations had no role in study design.

**Competing interests:** The authors have declared that no competing interests exist.

physiological data, such as cardiac and respiratory activities. Moreover, studies in social neuroscience suggest that synchrony during caregiving interactions—like physiological synchrony—is closely tied to affect regulation and stress management [2]. McFarland et al. [3] identified temporal clustering of vocalization, respiratory synchrony, and attention-related cardiac activity in interactions between children and mothers, highlighting a reciprocal influence between physiological and behavioral coordination on social interaction outcomes. Similarly, Busuito et al. [4] investigated skin conductance, respiratory sinus arrhythmia, and heart period synchrony in infants and mothers, and demonstrated connections to autonomic arousal and parasympathetic functions. Furthermore, the patterns between mothers and children during interactions have been proven to be closely related to parenting stress. For example, a study by Azhari et al. [5] shows that parent-child dyads with greater parenting stress exhibit different synchronized patterns in different brain areas. Woody et al. [6] explored the moderated effect of maternal history of major depressive disorder and the synchrony of physiological activity during mother-child interactions.

However, previous studies have focused on short-term interactive activities such as free play or shared task sessions [4,5,7]. They overlooked broader temporal interactions in daily life. Mothers and children share a living environment and interact from day to night, particularly in their physiological states. Unlike short-term interactions, the interactive patterns in such long-timescale signals, such as circadian rhythms [8], which offer critical insights into the body's natural regulatory patterns may also be closely related to caregiving stress.

In the current study, we acknowledge the importance of global information, such as circadian rhythms, by collecting daylong-24-hour electrocardiogram (ECG) data from children and their mothers in natural home environments. We aimed to explore the relationships between the long timescale child-caregiver interactions in the natural daily life of their physiological activities, such as ECG signals and parenting stress. We focused on the autonomic nervous system (ANS) activities of children and their mothers extracted from ECG data, which are closely related to mental stress [9]. Various indicators measure ANS activities, with heart rate variability (HRV) being the most frequently reported [10]. The intricate mechanism of the heart rate regulation system involves nonlinear feed-forward and feedback inputs, with nonlinear information significantly linked to functional state changes [11,12]. Using 24-hour at-home data allows for the investigation of authentic physiological interactions within natural environments, capturing inherent circadian variability and stress resilience in long-term physiological interactions not observable in shorter or laboratory-based studies. Furthermore, challenges arise in extracting, visualizing, and quantifying the wealth of information with nonlinear patterns within the data to facilitate comprehensive predictions, especially over a long timescale.

A recurrence plot (RP) is a tool used to analyze and visualize the patterns and dynamics of nonlinear signals. George et al. [13] have used it to quantify the circadian rhythm of the actigraphy data for distinguishing depressed and non-depressed populations. Li et al. [14] have proposed the use of RP to analyze and quantify the 24-hour ANS activities of the children and their mothers individually. However, RPs

can only quantify the patterns of a single signal without considering the interactive patterns between multiple resources. Thus, in the current study, we introduced an extended method, the cross-recurrent plot (CRP) [15], to visualize the dynamics of interactive patterns between two input signals. Subsequently, the generated CRPs serve as inputs to an unsupervised neural network model, the self-organizing map (SOM) [16], to decrease the dimensions of the input images and obtain useful representations. Multiple regressions were then used to assess the relationship between the extracted representations and mental stress.

The primary objective of this study was to explore the association between childcare stress and long-term ANS interactions between mothers and children. Existing research has neglected the analysis of prolonged daily interactions, particularly those involving circadian rhythms, and has often failed to account for the inherent nonlinear dynamics of ANS regulations. By employing the CRPs, which effectively capture these nonlinear interactions, this study directly addresses these gaps.

The comprehensive framework of this study is shown in Fig 1. The abbreviations used in this study are provided in S1 Table in S1 File.

## Related works

The application of RPs has proven instrumental in comprehending diverse physiological processes. In 1994, Webber and Zbilut [17] defined five nonlinear variables derived from RPs that were diagnostically useful for quantitatively assessing physiological systems and states. Subsequently, numerous studies have used RPs to measure respiratory and cardiac activities [18–21]. Previous studies have focused on mental state prediction. For example, Elalamy et al. [22] employed RP and transfer learning of physiological signals for emotion recognition. Ledesma-Ramírez et al. [23] found statistically significant changes in the quantification results from the RP of electroencephalogram signals between the rest and mental states of the brain. Sharif and Khan [24] compared mental stress between two groups of subjects, who are/are not examinees, using RP to measure the ECG signals.

As an extension of the RP, the CRP can measure nonlinear recurrent patterns between two signal sources, making it applicable to socially interactive contexts. Recent studies have extended CRP methods to account for time series of unequal lengths, improving their applicability to naturalistic data collection settings where minor discrepancies in sampling length or completeness are common [25], which extended the applications of CRPs to long-term data with complex dependencies across systems. CRP can be utilized to visualize and quantify coordination or synchrony in socially

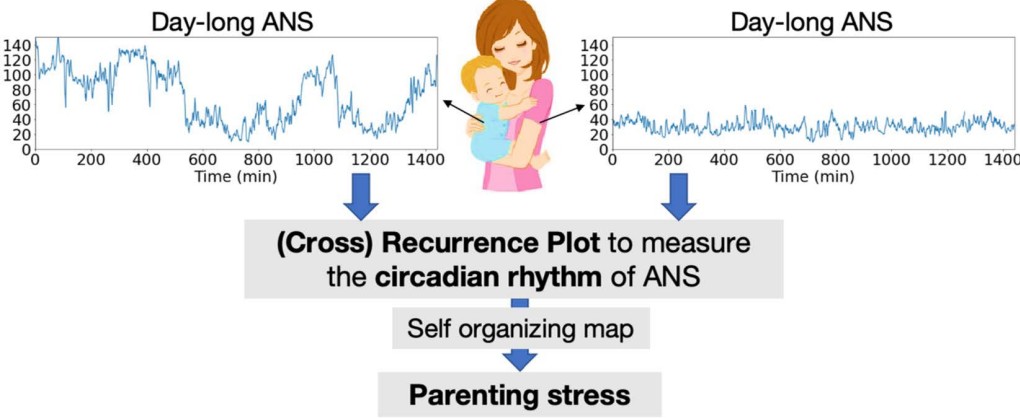

**Fig 1. Overview of the Proposed Methods.**

engaging partners that span various modalities, including behavioral and physiological studies [26]. For example, Veerabhadrappa, Hettiarachchi, and Bhatti [27] applied CRP to body motion, eye tracking, and HRV data to measure synchrony and predict social-physiological compliance in team workers. Adam, Mohammed, and Hui [28] explored the utilization of CRP for pattern matching of multidimensional physiological signals, classifying healthy and cardiac disease populations.

Researchers have also used CRP to study child-caregiver interactions. Xu and Yu [29] proposed a CRP-based representation to quantify the joint activities between children and their caregivers, revealing developmental changes in coordinative behavioral patterns. In Li at el. [30], CRP was employed to measure vocal coordination between children and their mothers in day-long audio recordings. Numerous studies have considered different modality data, such as co-movements [31,32], co-visualization [33], and motion regulation measured by gaze, voice, and motor [34]. However, compared to using CRP in behavioral data, using that in physiological data in the natural context of child-caregiver interactions has been scarce [35]. The development of measurement and assessment systems that focus on physiological indicators is extremely important because daily, especially long-term, physiological data monitoring is useful for early detection and prevention of mental health risks.

Research on the utilization of RPs or CRPs to analyze child-caregiver interactions is limited. Existing research often overlooks the broader temporal context, neglecting information embedded in global timescales such as circadian rhythms. Circadian rhythms play an important role in regulating various physiological processes such as hormone secretion, body temperature, and sleep-wake cycles [36,37]. These rhythms exhibit intricate patterns over the course of the day, influencing an individual's overall well-being and stress resilience. CRPs are uniquely positioned to capture interactive patterns between physiological signals, which are crucial for understanding the nuanced dynamics of stress within interpersonal interactions. The interplay between circadian rhythms and social interactions is complex and extends beyond linear relationships. CRPs provide a robust framework for unraveling these intricate connections, allowing a holistic examination of the influence of circadian rhythms on social dynamics.

## Data and ANS indices

**Data collection.**  One to three continuous days of ECG data were collected from 46 typical development children aged 36–48 months (41 ± 3 months) and their mothers, using Heartenote®, with a sampling rate of 256 Hz (Medical Device Certification Number 302 ACBZX00015000, JSR Corporation). Data were collected from the children's homes, ensuring a naturalistic setting without the presence of experimenters. ANS indices were extracted from the ECG data for subsequent analysis. Subjective parenting stress risk was assessed using the Japanese version of the Parenting Stress Index (PSI) [38,39] through a questionnaire administered to mothers once during the study. Participants were asked to perform ECG measurements and questionnaires within 1–2 weeks of the arrival of the measurement kit. We could not obtain data on what day of the month each subject performed the PSI. The PSI comprised 78 questions, each rated on a scale of 1–5 points. Higher scores for negative questions indicated elevated stress levels, whereas the scores for positive questions were reversed. The total points were used to measure the mothers' overall parenting stress, with a cut-off score of 221. A score of 221 points or above indicated that stress was excessively high, and that specialist support or intervention was required, which is the global standard set by the PSI questionnaire.

The PSI scores ranged from 105 to 284 (Fig 2). While most participants fell within the low-risk category for parenting stress (i.e., PSI < 221), their stress experiences varied across different levels. The primary objective of this study was not solely to predict high- or low-risk parenting stress but to leverage nonlinear tools, such as RP and CRP, to measure and quantify individualized ANS rhythms. This approach aimed to predict the PSI for each subject, providing a nuanced understanding of parenting stress beyond traditional risk categorizations.

This study was approved by The Medical Ethics Committee of Kyoto University (no. R2732) and Kyoto University Graduate School of Medicine, Faculty of Medicine and Kyoto University Hospital Medical Ethics Committee (no. R4015). The

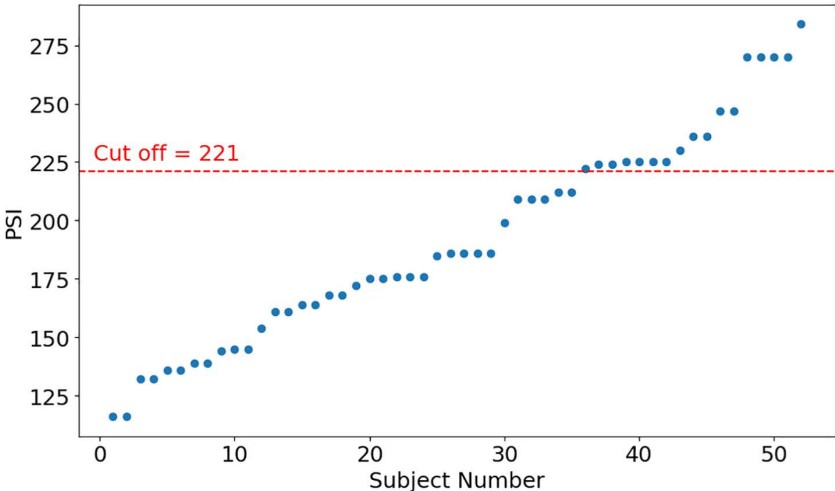

**Fig 2. The Distribution of PSI.**

participants were recruited from August 8th, 2022 to March 31st, 2023. Data collection was completed in the same period. Each caregiver participant was contacted by phone and provided a brief overview of the study before the experiment. For those participants who said they were willing to be sent information materials about the study, the research description, consent form, and consent withdrawal form were mailed to their homes. After reading the documents and understanding the study, if the participant agreed to participate in the study, they signed the consent document and returned it to Kyoto University. Participants were confirmed when the written informed consent form on paper was received at Kyoto University. The consent of the children participants was obtained from their caregivers.

### Indices measuring ANS activities

Three ANS activity-related indices were derived from the ECG data: the Standard Deviation of N-N Intervals (SDNN), the Cardiac Sympathetic Index (CSI), and the Cardiac Vagal Index (CVI).

The SDNN—a measure of the standard deviation of N-N intervals—was obtained by filtering artifacts and noise from the R–R intervals in the heartbeat data. This reflected the overall HRV, as shown in Fig 3 (left). The CSI serves as an indicator of sympathetic nervous system activity, whereas the CVI serves as an indicator of parasympathetic nervous system activity, as illustrated in Fig 3 (right). Both the CSI and CVI were calculated from the Lorenz plot [40] generated by plotting each R-R interval time length ($I_k$) against the subsequent interval time length ($I_{k+1}$) for a limited number of R-R intervals ($k$). Sympathetic and parasympathetic toni were estimated from the Lorenz plot, denoted as $L/T$ for the CSI and $log_{10}(L \times T)$ for the CVI, where T and L represent the lengths of the (inverse) diagonal, respectively. In this study, a window with a width of 300 s and a grid interval of 60 s was employed to extract the CSI and CVI. Datasets with artifacts exceeding 5% were excluded from the analysis.

Mental stress significantly influences ANS activities that are measured by HRV indices, including decreased SDNN and CVI, and increased CSI [41]. This is because mental stress is activated and influenced by the sympathetic branch of the ANS, leading to an elevated heart rate and reduced variability between consecutive heartbeats. The ANS activity exhibits nonlinear rhythms and plays an essential role in regulating various bodily functions. This regulation is governed by two branches of the autonomic nervous system—the sympathetic and parasympathetic nervous systems—which maintain a delicate balance known as the autonomic balance. The interactions between these branches are nonlinear—a characteristic often manifested in the observed recurrence patterns within the ANS.

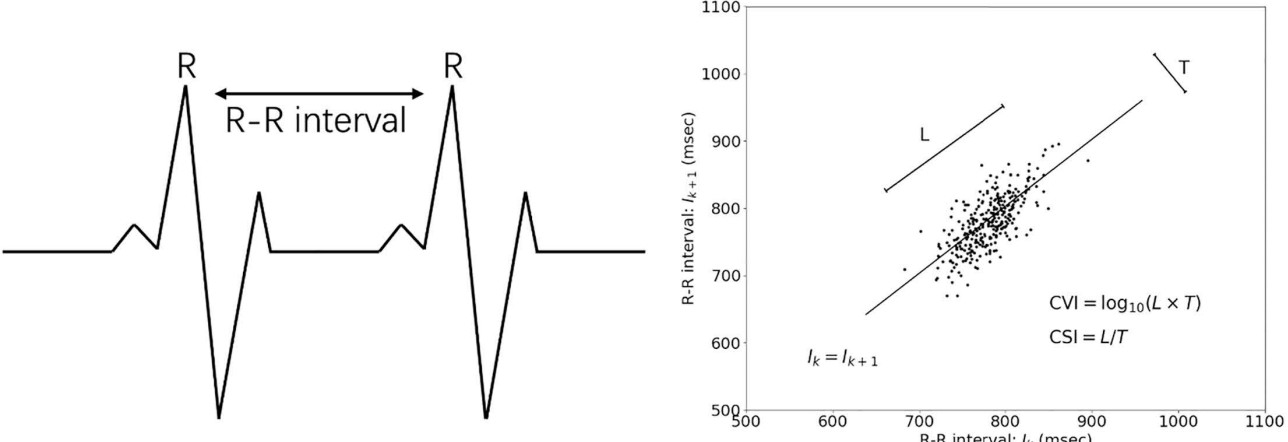

**Fig 3. ANS-related Indices Used in This Study, including SDNN, CSI, and CVI.** R-R intervals in heartbeats. (left) The NN interval represents the filtered R-R interval obtained by removing the artifacts and noise. Lorenz plots for calculating CSI and CVI [43]. (right).

Furthermore, the ANS is subject to the influence of a myriad of internal and external factors, such as stress and sleep, resulting in alterations in HRV recurrence patterns [42]. For example, stress can induce an increase in sympathetic activity, thereby disrupting the regularity of recurrence patterns. Sleep introduces day-night differences in ANS rhythms, with daytime activities affecting ANS activity during the night.

### Related techniques

Day-long ANS data contained a wealth of information, so we incorporated CRPs to visualize and quantify the interactive patterns exhibited by the ANS activities of the mothers and their children. CRPs were selected because they are specifically designed to visualize and quantify nonlinear interdependencies and synchronization between two physiological signals. The nonlinear dynamics refer to the complexity and irregularity in time series signals that result from the intricate feedback mechanisms regulating heart rate variability, which linear methods cannot adequately describe.

We first used SOM to downsize the dimensions of the input data, that is, the RGB data of the CRPs to quantify the rich and complex information on the CRPs. Then, we used the low-dimensional representations obtained from the SOM to predict the PSI. As CRP is an extended technology of RP, we first described the RP technology before investigating the utilization of CRP for a natural illustration.

### Recurrence plot

RP is used to analyze and represent patterns and relationships in time-series data, particularly in the context of nonlinear and complex systems. It is commonly used in physics, engineering, biology, and neuroscience, to study the dynamics and interactions within systems.

An RP is a graphical representation of the recurrence of a data point in a time series. In other words, it visualizes the times at which the same or similar values occur again in the same time series. A traditional RP converts the time-series data into a binary matrix where the elements are either 1 (indicating recurrence) or 0 (indicating no recurrence). The diagonal elements of the matrix represent self-recurrence, and the off-diagonal elements represent the recurrence of data points. Fig 4 (left) shows an example of an RP created by a sine sequence with $n$ data points, $X = [x_1, x_2, \ldots, x_t, \ldots, x_n]$ represents a time point. The examined RP shows the recurrent patterns of a 12-second sine signal with a frequency of 0.4

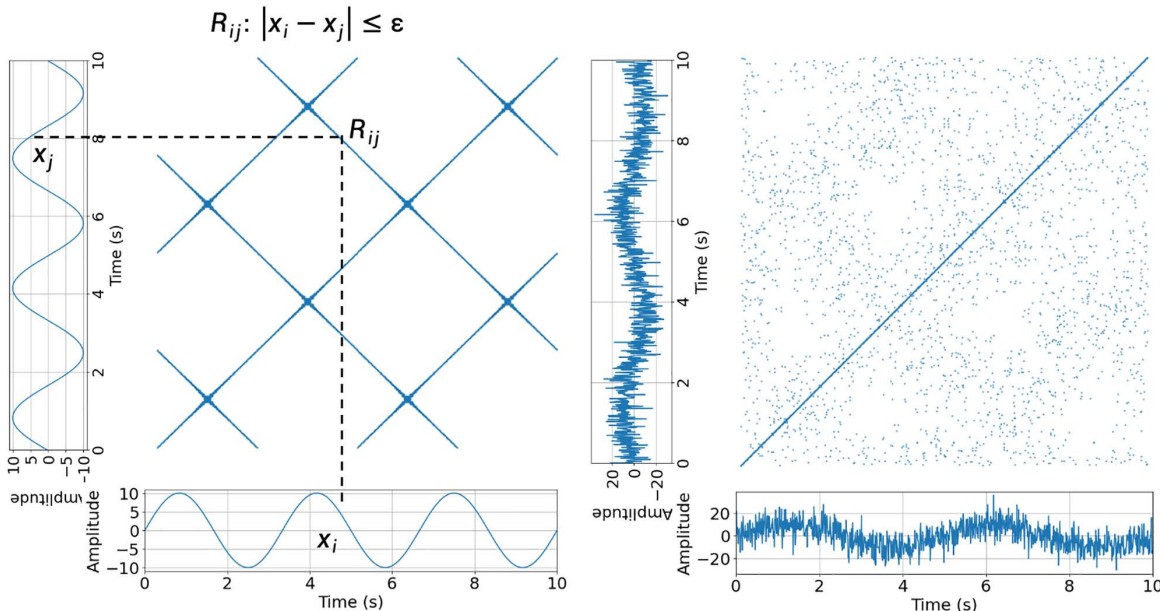

$$R_{ij}: |x_i - x_j| \leq \varepsilon$$

**Fig 4. The RPs Created by Sine Signals.** RP of a 10-second sine signal with a frequency of 0.4 Hz. (left). RP of a sine signal with the same length and frequency with added white noise (right).

Hz, made up by the recurrences $R_{ij}$, when $|x_i - x_j| \leq \varepsilon$, where the $x_i$ and $x_j$ are the values of the time series at time point $i$ and $j$, and $\varepsilon$ is a tolerance defined by the user.

The diagonal lines on the RP indicate the periods when the system remained in a similar state. Moreover, the clusters or structures on the plot present the patterns of recurrence that indicate specific behaviors or dynamics in the input data. For example, Fig 4 shows two sine signals with the same frequency. However, the signal on the right figure adds white noise, making the periodic patterns on the left figure more difficult to identify.

## Cross recurrence plot

CRP extends the concept of RP to analyze the similarity between two different time series across different time points. It shows the recurrence between corresponding points in the two time series, providing insights into the synchronization, coupling, and interactions between the two systems. CRP is particularly useful for studying interactions in complex systems. Fig 5 shows a CRP constructed using two sine sequences with different frequencies, for example, $X = [x_1, x_2, \ldots, x_t, \ldots, x_n]$. The CRP in the example presents the synchrony between $X$ and $Y$ by showing the recurrence $R_{ij}$ created when $|x_i - x_j| \leq \varepsilon$, where the $x_i$ and $y_j$ are the values of the two time series at time point $i$ and $j$, and $\varepsilon$ is a tolerance defined by the user.

Like RP, the diagonal lines in CRP represent synchronized or similar behaviors between the two systems, and the off-diagonal elements indicate the presence of non-synchronized or different behaviors between the two systems. As shown in Fig 5, the two sine signals were synchronized from 0 to 10 s, whereas the latter parts were asynchronized. Thus, different patterns were observed for CRP levels.

Both the RP and CRP provide a visual representation of the underlying dynamics of complex systems and can reveal important information about patterns, periodicities, and interactions that may not be apparent through traditional time-series analysis techniques. Particularly, CRPs visually quantify how two separate physiological signals simultaneously exhibit similar or synchronous nonlinear dynamics, reflecting complex physiological feedback loops that regulate the autonomic nervous system, which linear statistics fail to capture.

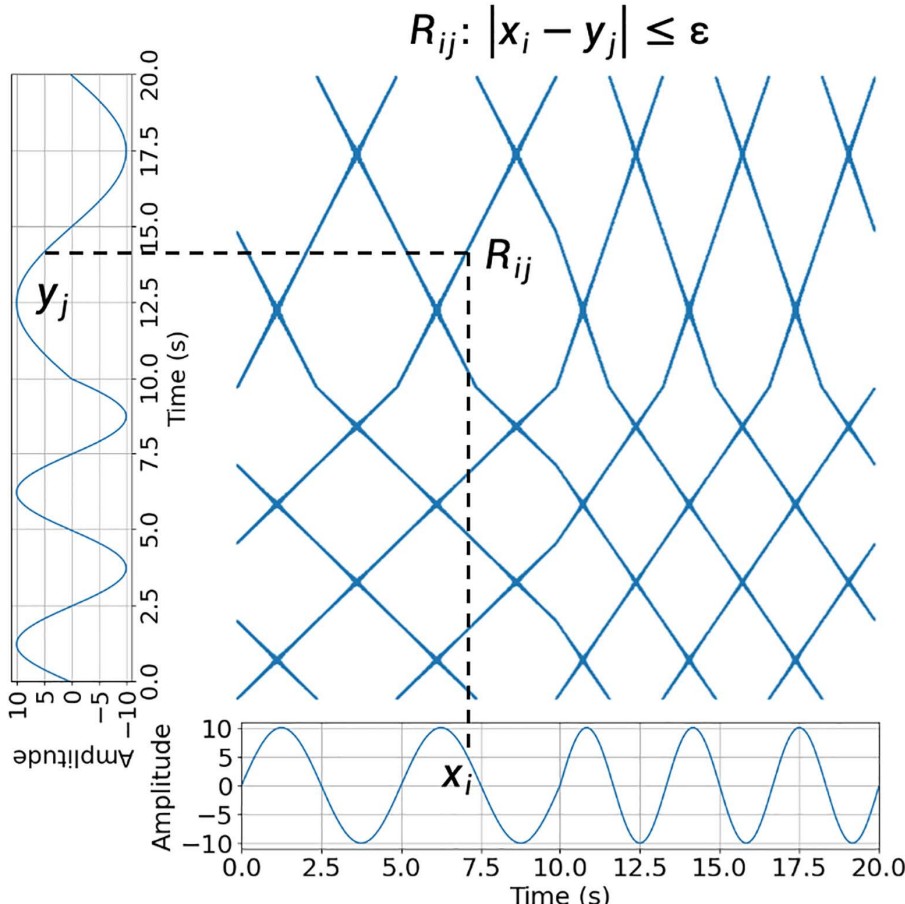

**Fig 5. A CRP of Two Signals That Synchronized Between 0–10 Seconds and Asynchronized Between 10–20 Seconds.**

## Self-organizing map

An SOM is an unsupervised neural network model for producing a low-dimensional expression from high-dimensional data. The goal of training an SOM is to represent an input space with high dimensions as a map space with low (typically two) dimensions. A map space is constructed by "nodes" or "neurons," which are arranged as a hexagonal or rectangular grid in a 2-dimension case. Each node in the map space is associated with a "weight" vector, which is the position of the node in the input space. Training is the procedure for updating weights to minimize their distances to the input data with the remaining topology induced from the map space. During training, the SOM adjusts the weight of each neuron based on the similarity between its weight vector and the input data. The result of this process is a map in which similar inputs are grouped in the same area of the grid. The concept of neuron distribution used in this study is presented in Fig 6.

## CRPs for ANS data

In this study, CRPs were constructed using dyad data to explore the interaction of circadian rhythms between the two parties. This approach allows for a nuanced examination of the coordinated patterns in the circadian fluctuations of the ANS within the dyadic interactions.

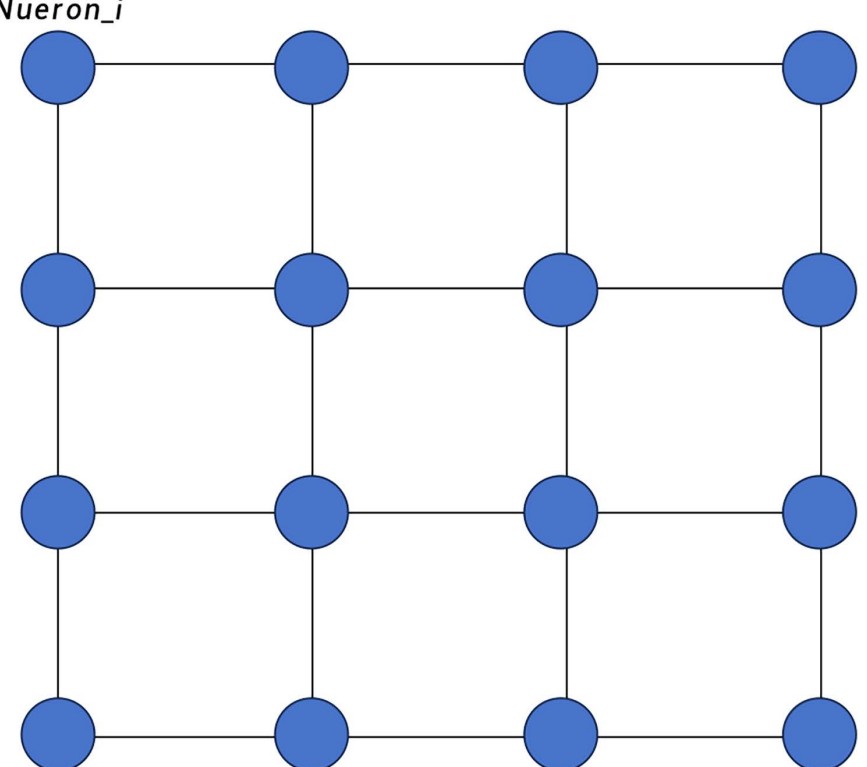

*Nueron_i*

**Fig 6. The Concept Figure of the Neuron Distribution of the SOM Used in This Study.**

## Data pre-processing

The input data for CRPs underwent the following pre-processing procedures:

1) Temporal Scaling: The data were segmented into a 24-hour scale, starting at 10 pm. This segmentation allowed the analysis to be conducted on a daily scale, facilitating the measurement of circadian rhythms, specifically the rest-awake cycle. Data shorter than 24 hours were excluded from the dataset.

2) Handling of Missing Data: Days with missing data exceeding 20% of the total length were excluded from the analysis. For days with less than 20% missing data, the gaps were filled using the mean values derived from the remaining data. Missing data attributed to device or manual errors of up to 20% were considered unsuitable for this study.

3) Cross-subject Normalization: A normalization process was implemented to account for individual differences in the fluctuation range of the signal across subjects.

4) Parameter Setting: Due to a lack of prior knowledge to determine the tolerance parameter ($\varepsilon$), a simplified approach was adopted by setting $\varepsilon = 0$ for both RP and CRP productions. Additionally, the normalized data were discretized by rounding them to one digit, mitigating details to facilitate the generation of recurrences on the plot.

Following these preprocessing steps, a dataset comprising 57 days of data from 32 mothers and their children was utilized for the subsequent analysis, in which 9, 42, and 2 dyads had 1, 2, and 3 days of data, respectively.

## CRPs for measuring the ANS interaction of the dyadic data

The preprocessed ANS indices for the child-mother dyads were utilized to construct each CRP. In Fig 7 (right), the CVI data of the mother are plotted along the horizontal axis, whereas the child's data are plotted along the vertical axis. If the values of the input ANS indices from the mother and her child at time point $i$ are equal, the recurrence $R_{xy\_i}$ is visible on the plot, where $x$ and $y$ represent the time series from the two parties. The recurrences displayed on the CRPs illustrate the synchronization between the mothers and their children in terms of ANS indices.

For instance, in Fig 7 (right), it is evident that the circadian rhythm of the dyad's CVI shows clear day-night differences. This circadian rhythm on CRP results from the recurrence of the CVIs of the mother and her child, which show similar and clear patterns for both day and night (Fig 7, left and middle). Please note that the CRP is not created by the RPs but is directly produced using the CVI data from the two subjects. The RPs presented here provide an easier understanding of the recurrent circadian rhythms of the CRPs. The CRPs provide a visual representation of the coordinated dynamics in the circadian rhythms of ANS activities within the dyadic interaction.

## Self-organizing map models for extracting representations used for predicting the parenting stress

The CRPs were used as inputs to train the SOMs. We trained different SOMs for each input ANS index, that is, the SDNN, CSI, and CVI, from mothers and their children. The same hyperparameters were set for the maps, which had the shape of 4×4 nodes. Each SOM was trained 25 times with different initializations, and similar results were obtained. Only one result for each SOM was presented in this study.

Furthermore, all the neuron activities in each SOM were measured for each input CRP. Multiple regressions were then conducted between the neuron activities and the PSI using the function (1) to investigate the predictability of the proposed methods.

$$PSI_i \sim Activity\_indice_i \qquad (1)$$

where $PSI_i$ is the PSI from mother $i$ and $Activity\_indice_i$ represents the neuron activities measured from the SOM trained by and the CRPs of SDNN, CSI, or CVI from subject $i$. In this study, the neuron activities were

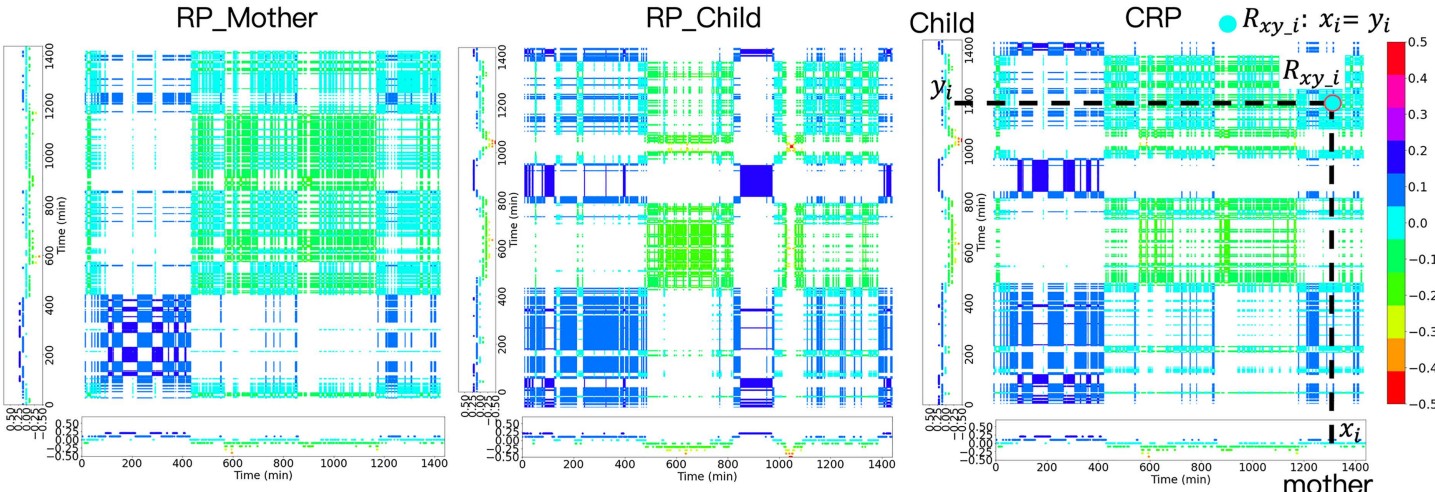

**Fig 7. The RPs Produced by The CVI Data of the Mother (Left) and Her Child (Middle). The CRP is produced by the dyad's CVI data (right).** The exaggerated light blue dot on the right figure shows the recurrence of the CVI data of the mother and her child at a point in time.

represented by the distance between the input CRP and the trained weights of each neuron. Thus, for each CRP $i$, $Activity\_indice_i = [d_1, d_2, \ldots, d_k, \ldots, d_{16}]$, where $d_k$ was calculated using the Euclidean distance between the input CRP and the trained weights of neuron $k$. The Root Mean Squared Error (RMSE), Mean Absolute Error (MAE), $R^2$, and the $p\_values$ were used to evaluate the efficiency of the multiple regression models.

## Results

### Different interactive patterns of ANS activities were observed among dyads

The interactive patterns of ANS indices between the two participants in a dyad were measured using CRPs. Figs 8 and 9 show the CRP results generated from the SDNN and the SOM, respectively. See S2 Figs in S1 File for the CRPs produced by the CSI and CVI data and their SOMs.

   The different patterns of recurrence of the mother-child ANS activities among the dyads are shown in Figs 8 and 9. After the SOM training, the CRPs were distributed in a 4×4 map, and each CRP was close to the neuron that had trained weights similar to its value. We assumed that the CRPs near the same neuron would have similar patterns; as a result, in Fig 9, only one represented CRP located at each neuron's location was presented. For ease of illustration, we focused on the example CRPs at the four corners of the trained SOM, that is, those in the red rectangle in Fig 9. The CRP in the upper right corner shows a clear circadian rhythm and good recurrence between the SDNN of the mother and her child. Compared to the right-upper CRP, the CRP at the bottom-right corner had fewer recurrences, that is, more blank areas appeared. This pattern indicates that the mother-child dyad shares little recurrence of SDNN data, either at night or during the day.

   Moreover, compared to the figures in the right panel, those in the left panel have more recurrent patterns but show unclear circadian rhythms, which indicates that neither the SDNN data of the mother nor the child showed clear day-night differences.

### Parenting stress predicted by the CRPs

Fig 10 (bottom panel) presents the results of multiple regression analyses of the CRPs using function (1). To ensure that the CRPs involving interactive patterns between the two partners were more effective at predicting parenting stress, we

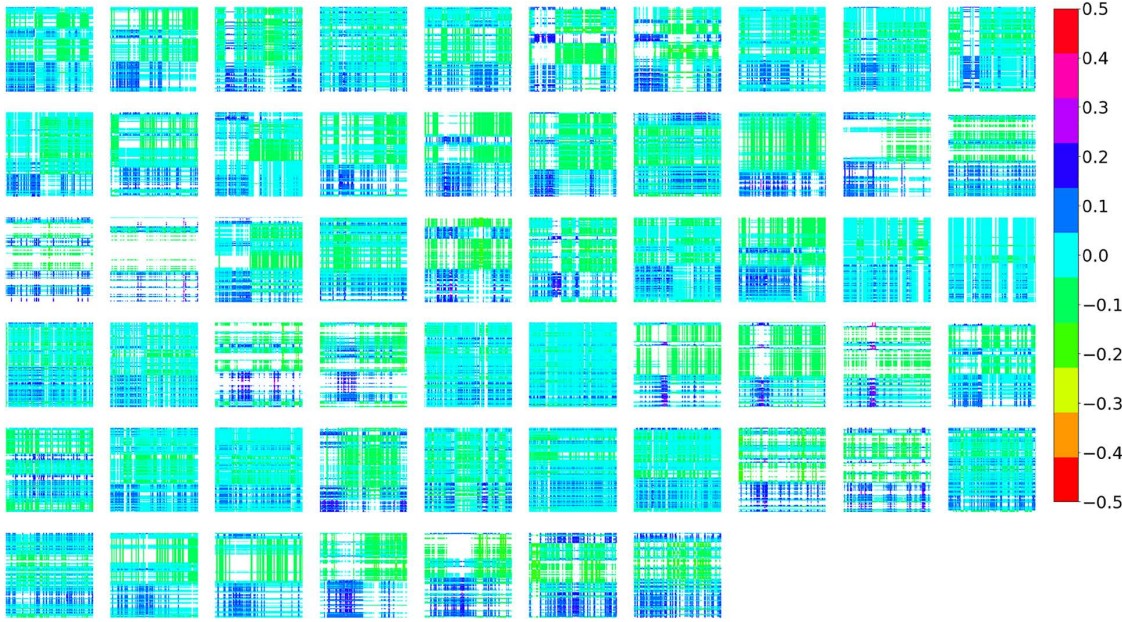

**Fig 8. CRPs Produced by SDNN Data.**

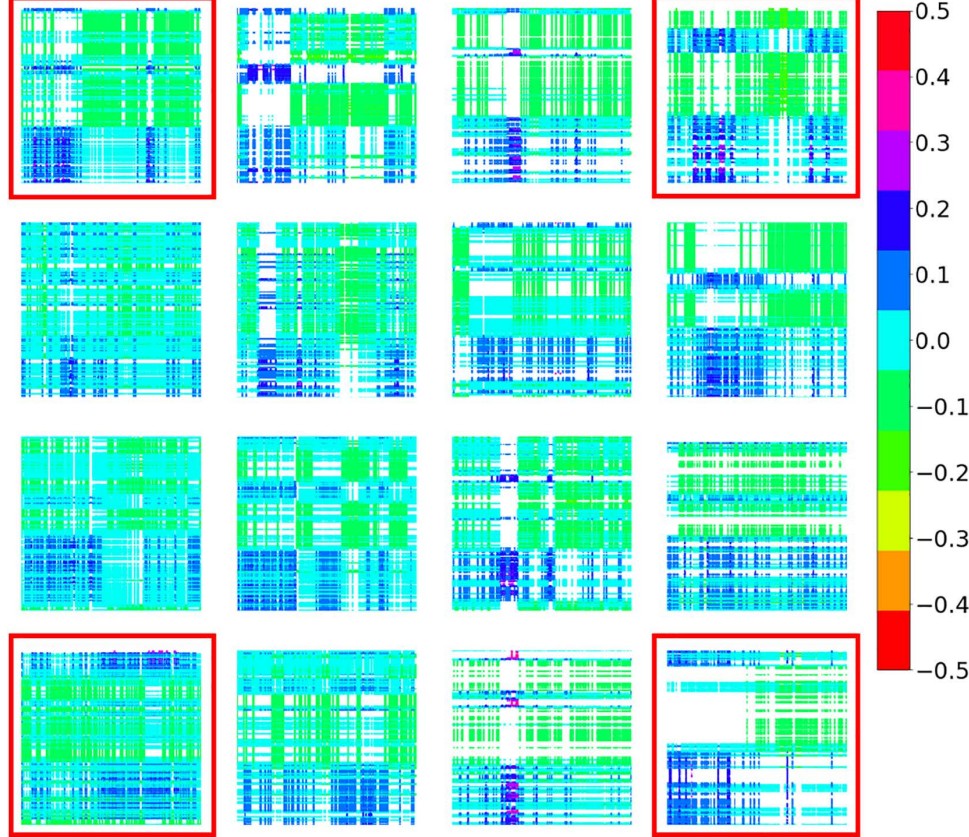

**Fig 9. SOM Result of the CRPs Produced by SDNN Data.**

created RPs using mothers' and their children's data separately. We then used them to train the SOMs and estimate multiple regressions between neuronal activities and PSIs. The results of the RPs are shown in the upper two panels of Fig 10. S3 Figs in S1 File presents the RPs and their SOM results.

The adjusted $R^2$, *f_pvalue*, *RMSE*, *and the MAE* were used to estimate the efficiency of the multiple regression models, which are presented at the top of each figure. A value of *f_pvalue* < .05 was assumed to be significant. The figures for the models with significant results are shown in green. The lower RMSE and MAE values indicate a better model fit. The model with the best fit is labeled in red. The red dashed line shows the ideal prediction; that is, the predicted value is equal to the actual value. The shadow represents the tolerance range, defined as the region where the predicted values are within one standard deviation ($\pm 1\sigma$) of the residuals around the actual values. More points located in the shadow and a shorter distance to the ideal line indicate a better prediction.

As presented in Fig 10, only one of the models from the RPs constructed by the ANS indices of the mother or the child significantly predicted PSI, whereas the CRPs from all three indices significantly predicted the PSI. Additionally, the best-fitting model was provided by the CRPs constructed from the mother-child dyadic CSI data, with the lowest RMSE and MAE values. These results show that, compared to the circadian rhythm quantified by the ANS indices from either the mother or the child alone, the interactive patterns between the partners better predicted the mothers' mental stress.

The presented study found that the CRPs, which quantify mother-child physiological interaction patterns, significantly outperformed individual RPs in predicting maternal parenting stress. Regression models using SOM-represented CRP

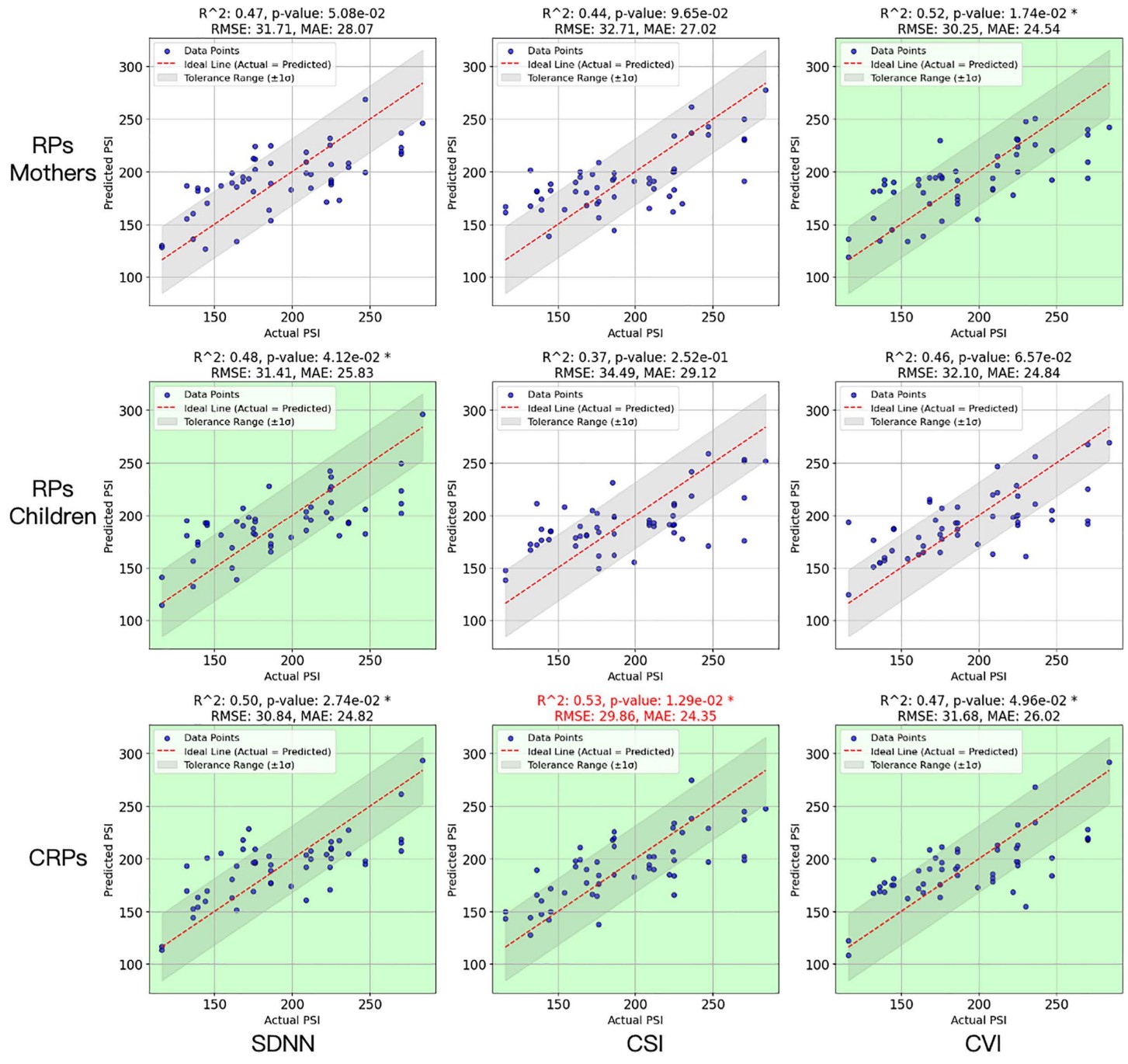

**Fig 10. Multiple Regression Results from RPs and CRPs.** * *f_pvalue* < .05.

features showed consistently higher explanatory power across all indices, with the strongest prediction from CRPs constructed using the CSI index. In contrast, only one RP-based model reached statistical significance, and it demonstrated lower predictive accuracy. These results suggest that measures of physiological synchrony and co-regulation, as

captured by CRPs, provide richer and more relevant information for assessing parental stress than individual physiological dynamics.

The CRPs were constructed by mapping the co-occurrence of similar physiological states between mother and child over 24 hours, capturing both within-day synchrony and transitions across circadian phases. In contrast, the RPs only mapped recurrence within a single individual's ANS signal, omitting interpersonal synchrony. Maternal stress emerges from the mother's internal state and the dynamic interaction between the caregiver and the child. CRPs reflect this dyadic physiological synchrony by capturing how two systems, i.e., mother and child, co-evolve over time. In contrast, RPs analyze only individual dynamics, ignoring the inter-personal coupling that is central to relational stress.

Moreover, the ANS signals are nonlinear, shaped by feedback loops, circadian modulation, and context-dependent adaption. When two non-linear systems interact, they often exhibit coupled oscillatory behavior. CRPs can detect patterns, including the joint recurrence structures, temporal lags and phase shifts, and the periodic synchrony between two systems, which are important information for predicting maternal stress and cannot be quantified by RPs. As a result, the CRP-derived models performed better than those using individual RPs.

## Discussion

Understanding the circadian rhythm of the ANS is crucial for gaining insights into the complex regulatory patterns that govern physiological processes. Traditional linear statistics, such as mean and standard deviation values, do not capture the intricate cyclicities inherent in ANS indices, which exhibit periodic and irregular patterns. The introduction of CRPs provides a nonlinear approach to efficiently measure these complex circadian patterns. This study employed CRPs to quantify the 24-hour ANS indices, revealing that nonlinear measurements, particularly those derived from circadian rhythm quantifications of RPs, outperform linear measurements in predicting maternal parenting stress. The heartbeat signals analyzed in this study involved dynamic mechanisms with various periodicities, emphasizing the need for nonlinear tools to unravel nuanced patterns within the ANS data.

Furthermore, circadian patterns, as exemplified by rest-active cycles, can differ significantly among individuals. This study corroborates findings from previous research [14] by demonstrating the significance of circadian patterns in predicting mental health states, such as stress. By showcasing the diversity of circadian patterns and their relevance to stress prediction, this study underscores the importance of utilizing CRP to discern subtle variations in ANS dynamics that linear statistics may overlook.

CRP is a useful tool for assessing the synchronization of circadian rhythms, particularly within mother-child dyads that share a living environment. The clarity of CRPs illustrates the complexities in day-night interactions of the ANS systems between the interactive partners, allowing for a nuanced examination of how individuals influence each other's physical and mental states. This study revealed that mothers and children exhibit varying degrees of synchronization in their circadian rhythms, with clear or unclear patterns discernible in the CRPs. The results demonstrate that parenting stress can be predicted more accurately using CRP results than conventional RPs, indicating that the clarity of circadian rhythm synchrony—as visualized through CRPs—offers a unique perspective on the co-regulation of the ANS within mother-child dyads. The observed associations between PSI and the CRP results emphasize the interplay between maternal mental states and the physiological interaction of circadian rhythms within the dyad. The capability of CRPs to measure this co-regulation suggests the potential for developing individualized child-rearing support methods tailored to diverse parents and children.

The use of SOM as an unsupervised neural network for dimensionality reduction in RPs presents a promising avenue for future research. While SOM effectively captures rich information from RPs, employing advanced models, such as deep convolutional neural networks tailored for image analysis, can enhance the accuracy of predictive models. Future studies should explore the potential of these sophisticated neural networks to gain deeper insights into the complex, high-dimensional data inherent in circadian rhythms.

The impact of circadian patterns on mental health and interpersonal dynamics warrants further investigation. Investigating the long-term effects of synchronized circadian rhythms within family units and developing intervention strategies based on CRPs could pave the way for innovative approaches. Additionally, investigations into the long-term effects of synchronized circadian rhythms in family units and the development of intervention strategies based on CRP could inform innovative approaches to improving mental and physical well-being in everyday situations. This field can advance our understanding of the intricate relationships between circadian rhythms, stress, and interpersonal dynamics by continually refining measurement techniques and leveraging advanced neural network models.

Although this study showed that circadian rhythms—especially the interactive patterns between mother and child dyads—have a close relationship with mothers' daily parenting stress, some limitations exist and need further exploration. For example, only physiological data were used to predict caregivers' mental stress. Behavioral data such as the sleep/wake cycles, physical contact between interactive partners, and environment-sharing conditions are necessary to identify the reasons for the distinct interactive patterns of ANS activities. In addition, there is variability in the 24-hour at-home data quality due to participant adherence to wearable device protocols, potential disruptions in normal activities, and signal artifacts from natural movements. Such noises will influence the quantifications of CRPs.

Considering these limitations, future research will integrate comprehensive behavioral assessments alongside physiological data, explore long-term developmental trajectories of circadian synchronization, and evaluate interactions designed to enhance the parents' caregiving life.

The relationship between the physiological and behavioral states is important for revealing the underlying mechanisms causing parental stress. The current findings will allow the implementation of public health policies and programs aiming at fostering the mental health of parents and caretakers who are normally exposed to the additional stress of child care. Potential real-world implications include the development of wearable physiological monitoring systems to alert caregivers about stress risks and targeted therapeutic strategies to improve caregiver-child physiological synchronization. Besides, these findings can guide the creation of specific biofeedback and personalized stress reduction programs that consider caregiver-child synchronization patterns.

## Conclusions

This study investigated the relationship between childcare stress and mother-child ANS interactions over a 24-hour period using a cross-recurrence plot. The results show that interactive patterns of ANS activities, particularly circadian rhythms, provide valuable insights into maternal parenting stress beyond what is achievable through RPs of individual ANS signals. The findings emphasize that the co-regulation of ANS activities within dyads significantly contributes to the understanding of stress in daily caregiving activities. CRPs effectively capture day-to-night differences and interactive patterns and underline the potential of CRPs as a tool for studying interpersonal dynamics and stress regulation in naturalistic settings. Understanding such a relationship between physiological and behavioral dynamics will facilitate the development of targeted interventions and public health strategies to support mental well-being among parents and caregivers.

## Supporting information

**S1 File. Appendix.** This file contains a table listing the abbreviations used in the paper and their meanings, as well as the complete set of recurrence plots and cross-recurrence plots generated for the analyses reported in the paper. (PDF)

## Author contributions

**Conceptualization:** Jiarui Li, Michiko Matsunaga, Masako Myowa.

**Data curation:** Michiko Matsunaga.

**Formal analysis:** Jiarui Li.

**Funding acquisition:** Jiarui Li, Michiko Matsunaga, Masako Myowa, Yukie Nagai.

**Investigation:** Michiko Matsunaga, Masako Myowa, Yukie Nagai.

**Methodology:** Jiarui Li.

**Project administration:** Masako Myowa.

**Software:** Jiarui Li.

**Supervision:** Masako Myowa, Yukie Nagai.

**Validation:** Jiarui Li.

**Visualization:** Jiarui Li.

**Writing – original draft:** Jiarui Li.

**Writing – review & editing:** Jiarui Li, Michiko Matsunaga, Masako Myowa, Yukie Nagai.

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
