## [Decision Letter · Decision Letter 0]

20 Apr 2025

Dear Dr. Li,

Thank you for submitting your manuscript to PLOS ONE. After careful consideration, we feel that it has merit but does not fully meet PLOS ONE’s publication criteria as it currently stands. Therefore, we invite you to submit a revised version of the manuscript that addresses the points raised during the review process.

We look forward to receiving your revised manuscript.

Kind regards,

Jerritta Selvaraj

Academic Editor

PLOS ONE

Journal Requirements:

2. Thank you for stating the following financial disclosure: This study was supported by a Grant-in-Aid for Scientific Research from the Japan Society for the Promotion of Science (JSPS) (17H01016, 19K21813, and 21H04981 to M. Myo.; 22K20314 and 24K16785 to J.L.); a Grant-in-Aid for JSPS Fellows (19J15173 and 22J01448 to M. M.); a grant from the Center of Innovation Program, Japan Science and Technology Agency (JPMJCE1307 to M. Myo., JPMJCR21P4 to Y. N.); a grant from the Moonshot R&D Program, Japan Science and Technology Agency(JPMJMS2296, JPMJMS2307 to M. Myo., JPMJMS2292 to Y. N.); a grant from the Next Generation Artificial Intelligence Research Center of The University of Tokyo (“Circadian Patterns Analysis on Autonomic Nervous System (ANS) data of Infants and Their Caregivers” to J.L.); a donation from the Hama Gakuen Corporation and the Shoei Kosan Corporation (“Childhood Dietary Habits and Gut Microbiome” to M. Myo.); and a donation from the Chouyoukai (“Dietary Habits and Intestinal Microbiota during Infancy” to M. Myo.).   

Reviewers' comments:

Reviewer's Responses to Questions

**Comments to the Author**

1. Is the manuscript technically sound, and do the data support the conclusions?

Reviewer #1: Partly

Reviewer #2: Yes

2. Has the statistical analysis been performed appropriately and rigorously?

Reviewer #1: No

Reviewer #2: Yes

3. Have the authors made all data underlying the findings in their manuscript fully available?

Reviewer #1: Yes

Reviewer #2: Yes

4. Is the manuscript presented in an intelligible fashion and written in standard English?

Reviewer #1: Yes

Reviewer #2: Yes

Reviewer #1: Authors made a review or critique of the study titled "Mother-child autonomic nervous system interaction as an indication of parental stress: 24-hour cross recurrence plot analysis.

These are the following research questions to answer by the authors

a. What is the primary objective of the study, and how does it address gaps in existing research on mother-child physiological interactions?

b. How does this study differ from previous research that focused on short-term, activity-based interactions?

c. What is the significance of using 24-hour at-home data for analyzing mother-child autonomic nervous system interactions?

d. Why were cross-recurrence plots chosen as the method for quantifying interactive patterns, and how do they capture the nonlinear dynamics of autonomic nervous system signals?

e. What key findings emerged from the study regarding the relationship between mother-child autonomic nervous system patterns and maternal parenting stress?

f. How do the day-to-night autonomic nervous system patterns, quantified through cross-recurrence plots, improve the prediction of maternal parenting stress compared to individual autonomic nervous system activity analysis?

g. What are the potential implications of these findings for understanding and addressing parental stress in real-world settings?

h. How might this research contribute to the development of interventions or tools for parents experiencing childcare stress?

i. What limitations might exist in using 24-hour at-home data and cross-recurrence plot analysis for this type of research?

j. What future research directions are suggested by the findings of this study, particularly in terms of exploring circadian rhythms and stress-related physiological interactions?

k. Are the concepts of "cross-recurrence plots" and "nonlinear dynamics of autonomic nervous system signals" clearly explained for readers unfamiliar with these methods?

l. How well does the abstract convey the complexity of the interplay between maternal and child autonomic nervous system activities?

m. What makes this study novel or significant in the field of parental stress and physiological interactions?

n. How does the study advance our understanding of the role of circadian rhythms in stress-related physiological interactions?

Reviewer #2: I would like to appreciate the work of the author. The paper looks good both technically and in standard quality.

Few suggestions can be included.

1. Improve the quality of the image as it looks blurred.

2. Most of the references are looking like older papers. Do check if they are really related papers.

**Do you want your identity to be public for this peer review?** For information about this choice, including consent withdrawal, please see our Privacy Policy

Reviewer #1: No

Reviewer #2: **Yes: ** Suganiya M

---

## [Author Response · Author response to Decision Letter 1]

1 Jun 2025

Dear Editor and Reviewers:

Thank you for considering our manuscript for potential publication in PLOS ONE and for providing us with useful advice and comments. We have modified the following points to meet the journal requirements:

1.Please ensure that your manuscript meets PLOS ONE's style requirements, including those for file naming. 

Response: We modified the formats of the manuscript, including the file naming.

2.Thank you for stating the following disclosure. Please state what role the funders took in the study. Please include this amended Role of Funder statement in your cover letter.

Response: We have added the states for the funding in the cover letter.

3.Please include captions for your Supporting Information files at the end of your manuscript, and update any in-text citations to match accordingly.

Response: We have included the captions for the supporting information files at the end of our manuscript and updated the in-text citations to match accordingly.

Additionally, we have modified the manuscript according to the suggestions given by the reviewers. Please check the response letter and the revised manuscript for the details.

Thanks again for considering our manuscript and please let us know if more modifications or information is needed.

Best regards,

Jiarui Li

---

## [Decision Letter · Decision Letter 1]

18 Jul 2025

Mother-child autonomic nervous system interaction as an indication of parental stress: 24-hour cross recurrence plot analysis

PONE-D-25-06809R1

Dear Dr. Li,

We’re pleased to inform you that your manuscript has been judged scientifically suitable for publication and will be formally accepted for publication once it meets all outstanding technical requirements.

Kind regards,

Jerritta Selvaraj

Academic Editor

PLOS ONE

Additional Editor Comments (optional):

Reviewers' comments:

Reviewer's Responses to Questions

**Comments to the Author**

Reviewer #1: All comments have been addressed

Reviewer #2: (No Response)

2. Is the manuscript technically sound, and do the data support the conclusions?

Reviewer #1: Yes

Reviewer #2: Yes

3. Has the statistical analysis been performed appropriately and rigorously?

Reviewer #1: Yes

Reviewer #2: Yes

4. Have the authors made all data underlying the findings in their manuscript fully available?

Reviewer #1: Yes

Reviewer #2: Yes

5. Is the manuscript presented in an intelligible fashion and written in standard English?

Reviewer #1: Yes

Reviewer #2: Yes

Reviewer #1: The authors have made the required modifications. The manuscript presents a novel and methodologically interesting investigation into mother-child autonomic nervous system (ANS) dynamics using 24-hour ECG recordings and cross recurrence plot (CRP) analysis. The focus on dyadic physiological coupling as an indicator of parental stress adds significant value to both developmental psychophysiology and family health monitoring domains.

Reviewer #2: I would like to appreciate the work of the author. The paper looks good both technically and in standard quality. Paper is accepted as all the comments have been responded.

**Do you want your identity to be public for this peer review?** For information about this choice, including consent withdrawal, please see our Privacy Policy

Reviewer #1: No

Reviewer #2: **Yes: ** Suganiya

---

## [Editor Report · Acceptance letter]

PONE-D-25-06809R1

PLOS ONE

Dear Dr. Li,

I'm pleased to inform you that your manuscript has been deemed suitable for publication in PLOS ONE. Congratulations! Your manuscript is now being handed over to our production team.

Kind regards,

on behalf of

Dr. Jerritta Selvaraj

Academic Editor

PLOS ONE